# Reintroductions of the Corsican Red Deer (*Cervus elaphus corsicanus*): Conservation Projects and Sanitary Risk

**DOI:** 10.3390/ani12080980

**Published:** 2022-04-11

**Authors:** Francesco Riga, Luciano Mandas, Nicola Putzu, Andrea Murgia

**Affiliations:** 1Italian Institute for Environmental Protection and Research (ISPRA), 00144 Rome, Italy; 2Agenzia Forestas, 09100 Cagliari, Italy; lmandas@forestas.it (L.M.); amurgia@forestas.it (A.M.); 3Agenzia Forestas, Actual Address Strada Fortino, 53, 14100 Asti, Italy; nicobiologo@gmail.com

**Keywords:** Corsican red deer, reintroduction, spatial behavior, species distribution model, Bluetongue virus

## Abstract

**Simple Summary:**

Reintroductions are important tools to achieve the long-term conservation of endangered species; however, these projects are dangerous for translocated animals. Reintroduced animals face novel habitats where food availability and secure resting sites are unknown, and in this context, such animals likely engage in long exploratory movements. Furthermore, reintroductions can be dangerous for domestic and wild animals inhabiting the reintroduction site due to the potential risk of catching diseases carried by reintroduced animals. Our study aimed to evaluate the spatial behavior of reintroduced Corsican red deer in central-eastern Sardinia and, on the basis of the obtained results, build a species distribution model to forecast the expansion of reintroduced deer and plan future reintroduction projects. Furthermore, we evaluated the sanitary risk for domestic animals (sheep) linked to Bluetongue virus potentially carried by deer. Our results highlighted the great habitat suitability of central–eastern Sardinia for Corsican red deer. However, deer are healthy carriers of Bluetongue virus, as confirmed by health analyses we performed on captured animals in the source areas. Particular attention in reintroduction planning is needed to avoid any negative impacts on domestic or wild animals of conservation measures.

**Abstract:**

The Corsican red deer is an endangered subspecies that needs artificial translocation projects to gain its complete recovery with the formation of viable, interconnected populations. Between 2007 and 2017, we performed two reintroduction projects in four sites in central–eastern Sardinia via tracking 32 deer by means of GPS/GSM radiotelemetry. On the basis of the obtained results, we built a species distribution model (SDM) using MaxEnt software, selecting 200 random points from the merged deer core areas as presence data. Furthermore, to evaluate the sanitary risk linked to artificial translocations, we analyzed deer positivity to Bluetongue virus (BTV) in the founder populations. The SDM showed a high deer capability to colonize central–eastern Sardinia, but it also showed the possibility of spreading BTV to domestic sheep because sanitary analyses confirmed the virus’ presence in the founder populations. Our main conclusion was that reintroductions are effective tools for the long-term conservation of the Corsican red deer, as long as sanitary risks are minimized by means of sanitary monitoring of translocated deer.

## 1. Introduction

Reintroductions and translocations are important in the conservation of endangered or locally extinct species and in cases in which natural recolonization is impossible, difficult, or requires a very long time [1]. In fact, the use of ex situ conservation measures, such as reintroductions, should be preferred to in situ intervention only in cases of necessity, when the achievement of conservation objectives is not obtainable otherwise [2,3]. There are numerous examples of deer reintroductions, such as: Persian fallow deer, i.e., *Dama mesopotamica* (Brooke, 1875), in Israel [4]; white-tailed deer, i.e., *Odocoileus virginianus* (Zimmermann, 1780), in the USA [5]; Père David’s deer, i.e., *Elaphurus davidianus*, in China (Milne-Edwards, 1866) [6]; and red deer, i.e., *Cervus elaphus* (Linnaeus, 1758), in Portugal [7]. However, a few studies have dealt with the post-release spatial behavior of cervids [8], with most observations on this topic available for Wapiti, i.e., *Cervus canadensis* (Erxleben, 1777) [9,10], and Persian fallow deer [4]. For the red deer, by contrast, post-release spatial data are scarce [11]. Knowledge of post-release dispersal and home range is indispensable because they are key factors affecting population viability [12,13,14]. The post-release spatial behavior of cervids is influenced by landscape spatial structures [11] and by human disturbances in the vicinity of the release area [15]. The movement and spread of reintroduced populations can be significant with a great area of occupancy [10] or very small with high release-site fidelity [14,16]. The presence of conspecifics near the release site should also be considered because many deer species show a high rate of social relationships [9].

In the middle of the twentieth century, the species distribution of *Cervus elaphus corsicanus* (Erxleben, 1777), a subspecies described in Corsica (France) and Sardinia (Italy) islands [17,18], was at risk of extinction as a result of habitat fragmentation, forest fires, legal hunting (in the past), and poaching (more recently) [19,20]. Over the last decades, through national and international protection measures (i.e., the inclusion in Appendix II of the Bern Convention and Annexes II and IV of the EU Habitats and Species Directives), awareness campaigns, and reintroductions [21], there has been an increase in both the number of animals and distribution range; therefore, the taxon is now considered as Least Concern in *The Italian Red List of Vertebrates* [22] and not listed by the IUCN [23]. Despite this, in Sardinia, the population of Corsican red deer remains fragmented and isolated. Furthermore, reintroducing the taxon was planned to allow population reestablishment in an isolated area where the species became extinct around 1950. In fact, two conservation projects, one funded by Forestas in 2009 and the other by the EU in 2013 (LIFE project, ’One Deer, Two Islands’), were carried out in the Ogliastra area (central–eastern Sardinia).

However, several issues should be considered from a health perspective when reintroduction projects are carried out. The risk of introducing wildlife diseases by animal translocation is increasingly important, involving wildlife, domestic animals, and human health [24]. In Sardinia, Bluetongue virus (BTV) is one of the most common ruminant diseases, caused by an RNA virus belonging to the *Orbivirus* genus (*Reoviridae*), transmitted by *Culicoides* spp., and affecting both wild and domestic animals (especially sheep) [25,26]. EU Regulation 2018/1882 lists the infection associated with BTV as a category C disease: relevant to some EU Member States and for which measures are needed to prevent it from spreading to parts of the Union that are officially disease-free [27].

Wild ruminants may play an important role as potential reservoirs of BTV and for its dissemination and persistence [28]; deer (similar to cows and goats) do not usually show any clinical signs of disease and can carry the virus and transmit it to other ruminants. BTV outbreaks and persistence are affected by climatic, environmental, and socioeconomic factors and sympatry with wild ungulates [29,30].

In this scenario, movements and home ranges of reintroduced Corsican red deer increase the risk of facilitating the spread of BTV, introducing it in the releasing site or allowing it to colonize newly suitable areas.

The aims of this study were: (1) to analyze home ranges and movements of reintroduced deer; (2) to provide a description of environmental variables determining deer dispersion; (3) to evaluate the dissemination risk of BTV caused by reintroduced Corsican red deer.

## 2. Materials and Methods

### 2.1. Study Area and Founder Populations

The reintroduction programs of Corsican red deer to Ogliastra were carried out from 2009 to 2019. Founder deer were captured in Costa Verde protected area, in a 250 ha fenced area in the municipality of Seui, and in a rearing compound in Sa Portisca area. Animals were baited into small enclosures and immobilized with 0.8–1 mL of a mixture of detomidine in combination with tiletamine and zolazepam, which was injected with the aid of a blowgun; anesthesia was reversed with atipamezole. This drug combination sedates deer without capture myopathy risk. After sedation, deer were hosted in stable boxes until the serological results (see Section 2.4).

Some free-ranging individuals were captured utilizing an off-road vehicle and a dart gun. In this case, animals were immobilized with xylazine mixture (Zoletil [31]) and injected via Pneudart (Pneudart Inc., Williamsport, PA, USA) equipped with a radio transmitter. The age of each animal was estimated by examining tooth eruption and wear, adopting the following age classes: yearling <12 months, subadult from 12 to 24 months, and adult >24 months.

Deer were released in 4 sites, where there were no free-ranging wild herds, and selected based on a feasibility study. These sites were in the subspecies’ historical range, within a protected area, and with suitable habitats available [32]. The four sites were (Figure 1):Montarbu (39.855852, 9.387882) is located at an altitude between 600 and 1350 m above sea level (a.s.l.). In this area, Mediterranean holm oak (*Quercus ilex* L.) forest is predominant but alternates with Mediterranean scrub and garigue in open areas. Eight deer were released (4 males and 4 females), all coming from the fenced area.Taccu (39.812584, 9.456521) is at an altitude between 700 and 1000 m a.s.l., with alternation among holm oak wood and reforestation of conifers (*Pinus* spp.) and Mediterranean scrub. Twenty-two animals were released (8 males, 14 females), among which 12 originated from the fenced area of Montarbu and the others from the Costa Verde wild population.Sa Portisca (40.172065, 9.528568) is located at an altitude between 300 and 1000 m a.s.l. There is a predominance of holm oak (*Quercus ilex*) and juniper (*Junipers* spp.); the shrub layer is characterized by *Pistacia lentiscus*, *Rhamnus alaternus*, *Phillyrea latifolia*, *Erica arborea*, and *Arbutus unedo.* In this area, 41 deer were released, with 17 coming from the wild population of Costa Verde (3 males and 14 females) and 24 from the Sa Portisca rearing compound (13 males and 11 females).Rio Nuxi (40.185772, 9.545929) is located in the most important Sardinian mountainous area (Gennargentu) and is up to 1800 m a.s.l.; predominant habitat is wooded matorral with *Juniperus* spp. and Mediterranean shrubs with spiny brooms (*Calycotome spinosa*). Thirty-six deer (13 males and 23 females) captured in the Costa Verde area were released here.

### 2.2. Radiotelemetry

In sites 1 and 2, translocations and release operations were conducted between January 2009 and February 2010, and 20 animals were equipped with GPS/GSM radio collars (model Tellus 12-channel). In sites 3 and 4, reintroduction took place between December 2013 and February 2019; 29 deer were equipped with Lotek GPS/GSM model GPS8000SGU. GPS collars were programmed to acquire six locations per day in sites 1 and 2 and two locations per day in sites 3 and 4. At the beginning of the project, an independent experiment was carried out in the Taccu site to determine the mean location error by placing collars in 39 test locations across the study habitat in 5 habitat types. For each test site, we calculated the standard deviation from the attempted location; we calculated the mean location error (±35.51 m) by averaging the standard deviations across the 39 test sites [33,34].

We analyzed data from GPS collars applied to 32 released deer. We excluded GPS collars with fewer than 90 days of data collection because they were considered insufficient to describe spatial behavior. A screening of locations according to DOP (dilution of precision) and the number of satellites used in the position acquisitions was carried out. Three-dimensional localizations (3D) with DOP > 10 and two-dimensional localizations (2D) with DOP > 5 were discarded [35,36,37].

GPS locations for translocated deer included in the analysis were stored in a spatial data frame in R Software [38]; 95% minimum convex polygons (MCPs) [39] were created utilizing adehabitatHR Package [40]. Furthermore, home ranges of deer were calculated using Brownian bridge movement models (BBMMs) [41], setting the GPS location error (σ_2_) to a radius of 35.51 m [34]; the Brownian motion variance parameter (σ_1_) was empirically estimated using the *liker* function [40]. The 95% BBMMs were calculated on the basis of a grid size of 100 × 100 m; the core area for each deer was calculated, extracting isopleths delimiting the area most intensively used by each deer [42,43]. Dispersal, the maximum distance reached by each animal with respect to the release point, was calculated for the period between release date and the end of the battery charge of every GPS collar. We investigated differences in dispersal and home range between sexes and between the four release sites by Kruskal–Wallis tests as the sample size was too small and unbalanced to perform multivariate analyses. The individual home ranges were converted to shapefile and mapped in QGis 22.2 [44].

### 2.3. Suitability Model

We developed a species distribution model (SDM) using the maximum entropy algorithm MaxEnt [45] to forecast the reintroduced Corsican red deer’s spatial expansion on the basis of dispersion and home range data in Sardinia.

#### 2.3.1. Tracked Animal Data

As our topic was the dispersion of reintroduced deer, we did not consider the presence data of free-living animals in south Sardinia natural populations, as in [32], but we used data obtained from the radiotelemetry study carried out in the four study areas, extracting 200 random points and setting 200 m as the nearest distance from points inside the area bounded by animals’ BBMM core areas.

#### 2.3.2. Environmental Variables

We selected a set of geographical predictor variables (GPVs), according to the species’ ecological requirements [32,46,47,48], deriving them from Sardinian Corine Land Cover (1:25,000) from a regional geographic information service (http://www.sardegnageoportale.it/, accessed on 22 February 2021), a digital terrain model (20 m), hydrographic maps from National Geoportal (http://www.pcn.minambiente.it/mattm/en/, accessed on 22 February 2021), and the principal road layer.

Land use typologies were reclassified into 11 variables (urban, wooded agriculture, managed woods, natural woods, natural pastures, maquis, extensive agriculture, intensive agriculture, beaches and dunes, natural areas with limited vegetation, and water). Furthermore, we used the forest fragmentation classes (core, isolated, perforated, edge) using the software GuidosToolbox [49,50]. Finally, we also considered the following variables: distance from core forests, distance from forests, distance from urban areas, distance from principal roads, acclivity, exposure, and Shannon diversity index using reclassified ecological variables with a circular radius of 200 m.

#### 2.3.3. Modeling Procedures

The Corsican red deer SDM was built using the default MaxEnt settings, 100 replicates, and a bootstrapping approach. The output was converted into ’suitable’ and ’unsuitable’ using the 10th percentile of probability occurrence as a threshold [51,52].

### 2.4. Deer Health Monitoring

To monitor the health status of Corsican red deer, from 2007 to 2017, blood samples of 94 captured deer (including deer that would be reintroduced) were collected from the jugular vein and tested for antibodies against BTV using a competitive ELISA assay [53]. Samples positive to BTV antibodies were tested by means of a virological RT-PCT test [53].

## 3. Results

### 3.1. Reintroduced and Tracked Deer

In total, 107 Corsican red deer were reintroduced to the four study areas, and 49 were equipped with GPS collars. Thirty-two of them were used to analyze dispersal behaviors and home ranges (Table 1); 17 deer were excluded due to collar failure or an insufficient tracking period.

### 3.2. Spatial Behavior

For all 32 deer, we estimated the total home range (MCP 95%, BBMM 95%, and BBMM core) and distance (Distance) from the release point (Table 2).

The medians (standard errors) of the variables were: MCP 95% = 671.31 ha (SE = 500.66); BBMM 95% = 834.81 ha (204.50); BBMM Core = 221.79 ha (64.94); and Dist = 4247.46 m (733.65).

No statistically significant difference resulted in home range sizes or dispersal distance when we compared among sex, age, and origin classes of the reintroduced deer.

However, taking into consideration the reintroduction sites, significant differences were observed for all studied variables (Table 3, Figure 2).

### 3.3. Species Distribution Model

Within the area delimited by the merged core areas of the Corsican red deer tracked in this study, we selected 200 random points and used them (considered as independent presence data) to perform the SDM using MaxEnt and to forecast the territorial expansion of the reintroduced deer.

The resulting distribution model showed a good predictive ability (AUC = 0.914 ± 0.015), and the binary map highlighted the good expansion capability of the reintroduced deer (Figure 3).

Table 4 shows the relative contributions of the considered variables. The main variables affecting the potential distribution of Corsican red deer were those related to anthropogenetic disturbance (distance from urban and from principal roads) and wood presence and fragmentation, with habitat suitability increasing at a greater distance from road and cities and with proximity with forest area and core (continuous) forests.

### 3.4. Health Monitoring

Twenty-six out of the ninety-four deer tested were positive for BTV antibodies (prevalence 28.8%); among them, seven were positive to the RT-PCR test (prevalence 7%). However, deer that were positive to the virological test (being an asymptomatic carrier of BTV) were excluded from reintroduction activities.

## 4. Discussion

The observed differences in spatial use of deer at reintroduction site-level probably depended on the morphological structure of site 4 (landscape, habitat composition, etc.). Consequently, we assumed that the core area selected by reintroduced deer was a good descriptor of space use as it was not affected by the sex, age, or origin of animals.

The home ranges observed in our study were larger than those previously found in the natural population of Monte Arcosu [46] and larger than those reported for the Spanish red deer subspecies, which occupies a similar Mediterranean environment [54]. In fact, in the Monte Arcosu area (southwestern Sardinia), the mean home range size (MCP 100%) was 122.1 ± 97.1 ha; males had an annual MCP 100% of 190.1 ± 100.5 ha, and females had an annual MCP 100% of 113 ± 92.6 ha [46]. In Monfragüe National Park, the annual home range size (MCP 100%) of Spanish red deer was, on average, 337.8 ± 184.8 ha (655.4 ha for a male deer and 258.1 ± 59 for four females) [54]. In other non-Mediterranean populations, red deer show greater home range sizes: 3600 ha for males and 840 ha for females in Bialowieża National Park [55]; 10,600–11,800 ha in a Sitka spruce plantation in Scotland [56]; and 1.582 ± 175 ha (kernel estimation of 95%) in a low-human-activity area in Saxony-Anhalt (Germany) [57]. However, the core area sizes were smaller (338.72 ± 367.37 ha), which reflected that deer may have settled in a stable area. In fact, reintroduced animals face new environments, and they should perform long exploratory movements in search of food and secure resting areas, resulting in home ranges several times larger than animals living in natural populations [58]. The maximum distance from the release site varied by 1.4 km–22 km, with higher values observed for M38, M40, and M43, two adult and subadult males reintroduced to study site 4 in Rio Nuxi. However, the observed movement size may have been affected not only by the ecological characteristics of the release site but also by the temperament and life history of the individuals [59,60].

Nevertheless, the aim of our work was to acquire information on the spatial behaviors of translocated Corsican red deer and build an occupancy model of the subspecies in Sardinia. From this point of view, spatial behavior variability among tracked deer may improve the model by making it more general.

The most suitable expansion area for *Cervus elaphus corsicanus*, as identified by our model, was in central–eastern Sardinia where the forest is continuous or where forest bridges connect small forest cores and within the maximum distance travelled by the tracked animals from the release sites and within the median size of the BBMM 95% home ranges. Furthermore, this region has very few urban settlements and a poorly developed road network.

We can foresee the complete colonization of this suitable area in western–central Sardinia in a medium–long period, depending on whether other reintroduction programs will be implemented quicker. These will greatly improve the long-term conservation of the Corsican red deer, ensuring the survival of wild populations and the colonization of the ancient distribution range of the subspecies.

However, this scenario may conflict with the need to limit the spreading of BTV among domestic sheep. As our results showed, Corsican red deer may play an important role as reservoirs of BTV and, as highlighted for other wild ungulates [28,61,62], in the dissemination and persistence of BTV in a habitat. This risk is particularly high in Sardinia, where BTV is widespread. In fact, domestic sheep amount to 3,036,666 heads (57.13% of the Italian grand total) [63], and the SDM we developed showed an overlap of deer-suitable area with the distribution of sheep (Figure 4). Bidirectional virus transmission (carried by *Culicoides* spp.) is probable, and more information on the interactions between wild and domestic ungulates needs to be acquired and included as predictors of BTV spreading in Sardinia [64].

## 5. Conclusions

Reintroduction projects for endangered taxa play a significant role in their conservation, especially when remaining populations consist of small numbers of animals confined in small and fragmented distributional ranges. The recent natural history of the Corsican red deer highlights a similar context. The subspecies disappeared from northern and central Sardinia in 1940 (and from Corsica in 1970), and the total number was reduced to individuals split into three separated populations; the recovery began in the 1980s when the taxon was subject to active conservation measures (i.e., ex situ conservation enclosure), specific protected areas were established, and more efficient poaching control systems were adopted [20,21,65]. However, the actual subspecies size (up to 10,000 heads based on Forestas’ unpublished data) depends mainly on the recovery of the three historical populations and secondarily on initial reintroduction programs. To assure long-term survival, a network of viable populations (i.e., a metapopulation) is needed. The SDM produced on the basis of the spatial behavior of tracked deer confirmed the actual availability of a suitable area for reintroductions.

To achieve the objective of a widespread reintroduction of wild populations of Corsican red deer, wildlife managers should also consider health items to prevent the diffusion of BTV from affecting the human attitude toward reintroduction projects. In particular, the following should be considered: (i) health monitoring of deer source areas and every founder individual; (ii) continuous health monitoring of the domestic sheep in release sites; (iii) vaccination of deer before translocation; (iv) assessing the presence of carrier insects; (v) information campaigns of projects; (vi) limiting reintroductions to southern Sardinia, where the density of sheep is the lowest.

## Figures and Tables

**Figure 1 animals-12-00980-f001:**
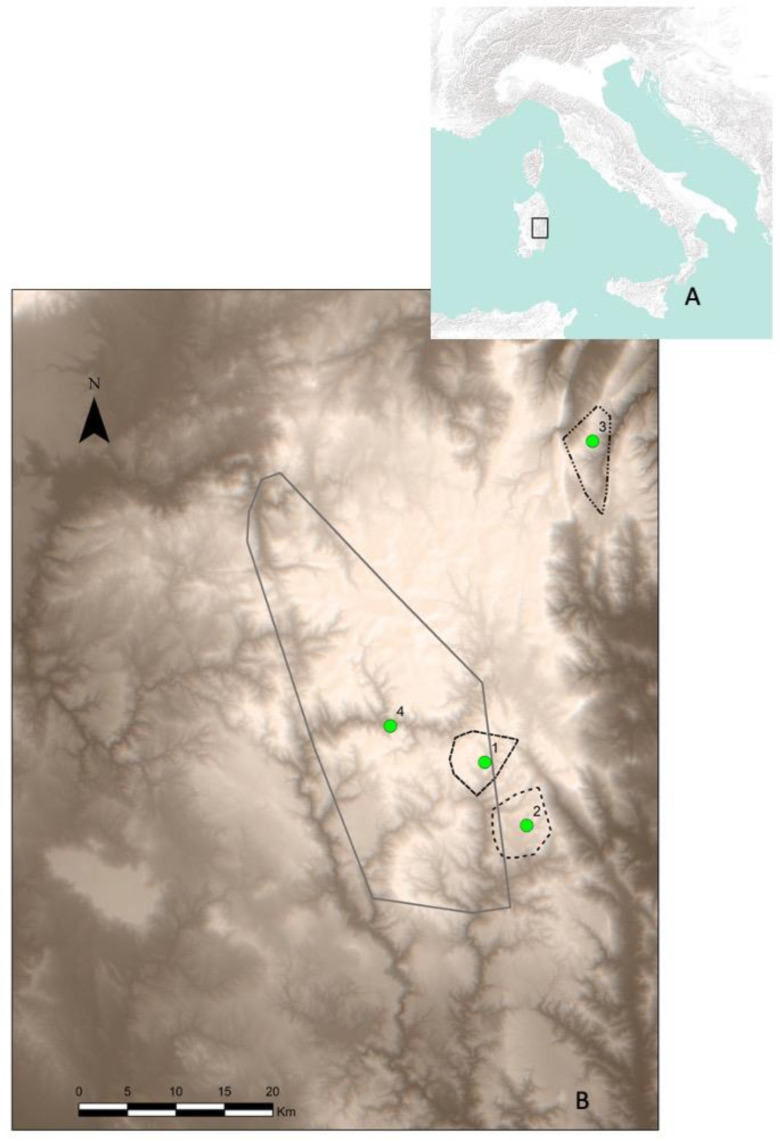
(**A**) Italy; (**B**) Sardinian study area: green dots are the reintroduction sites (1, Montarbu; 2, Taccu; 3, Sa Portisca; 4, Rio Nuxi); lines are the outermost locations of Corsican red deer tracked in this study.

**Figure 2 animals-12-00980-f002:**
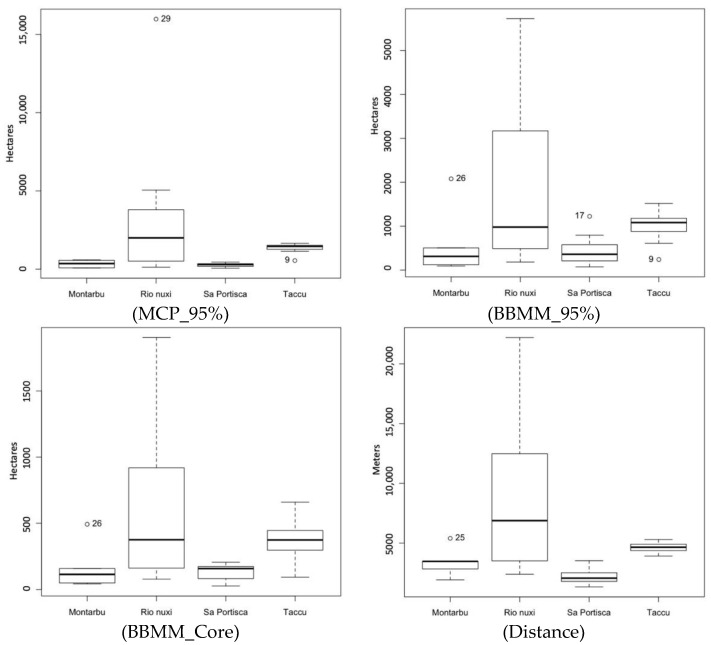
Box plots of home ranges and dispersal compared among reintroduction sites.

**Figure 3 animals-12-00980-f003:**
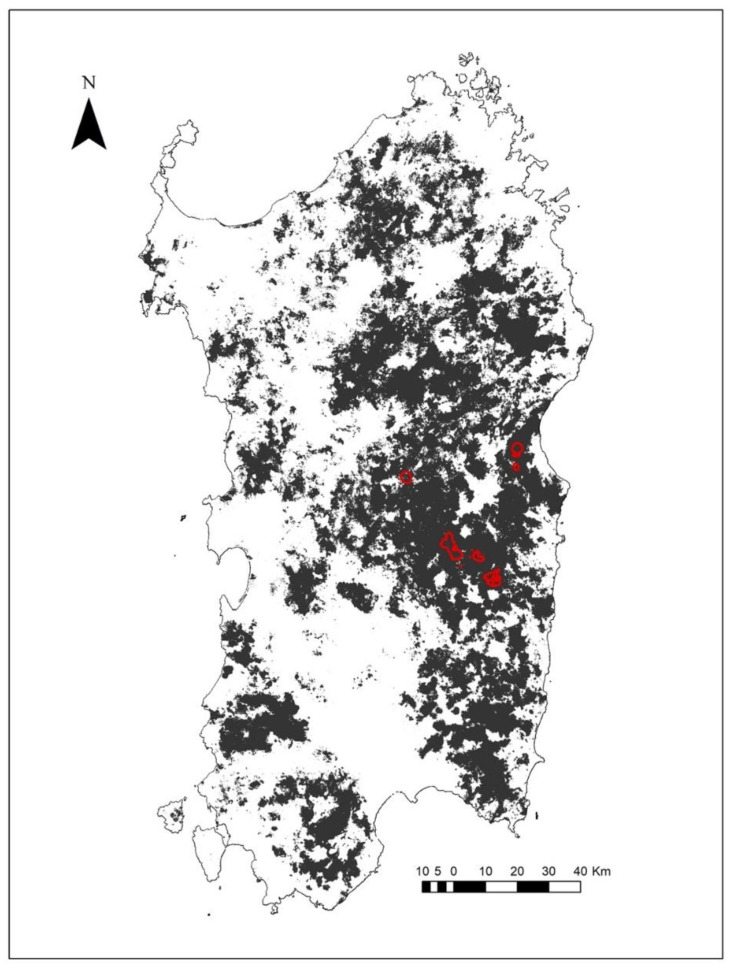
Binary species distribution model for the reintroduced Corsican red deer (the suitable areas are in black and the merged core areas are in red of tracked deer).

**Figure 4 animals-12-00980-f004:**
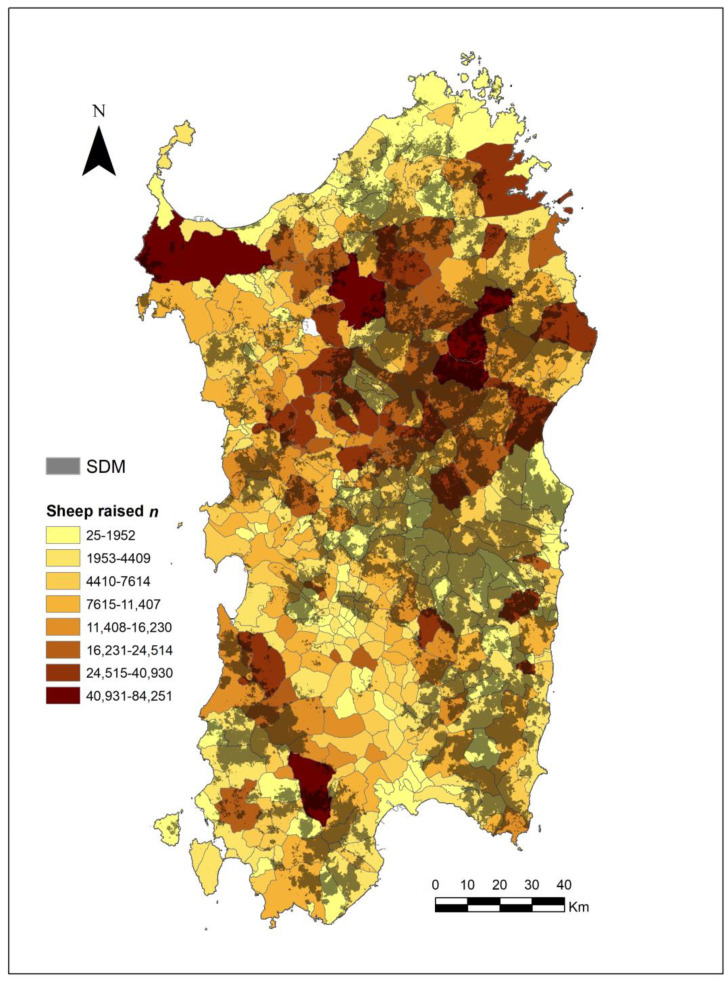
Corsican red deer SDM and distribution (at municipality level) and number of domestic sheep.

**Table 1 animals-12-00980-t001:** GPS-tracked Corsican red deer and main individual characteristics.

Deer	Sex	Age Class	RS	RD	RtDs	Origin
F01	F	AD	2	03/02/09	775	Captivity
F02	F	SA	2	14/02/09	653	Captivity
F04	F	SA	2	14/02/09	623	Captivity
F05	F	AD	2	14/02/09	507	Captivity
F07	F	SA	1	26/02/09	593	Captivity
F08	F	AD	1	26/02/09	568	Captivity
F12	F	SA	2	21/10/09	506	Wild
F13	F	AD	2	21/10/09	517	Wild
F14	F	AD	2	23/11/09	791	Wild
F17	F	AD	2	09/02/10	407	Wild
F23	F	AD	3	06/02/14	578	Captivity
F24	F	AD	3	06/02/14	509	Captivity
F30	F	AD	3	25/02/14	363	Wild
F41	F	AD	3	14/10/14	739	Wild
F42	F	AD	3	14/10/14	756	Wild
F43	F	AD	3	14/10/14	780	Wild
F54	F	AD	4	20/03/15	406	Wild
F55	F	SA	4	20/03/15	575	Wild
F56	F	SA	4	20/03/15	435	Wild
F64	F	AD	4	08/10/17	307	Wild
M01	M	AD	2	03/02/09	328	Captivity
M02	M	AD	2	03/02/09	584	Captivity
M04	M	AD	2	14/02/09	453	Captivity
M06	M	SA	1	26/02/09	458	Captivity
M07	M	SA	1	26/02/09	300	Captivity
M08	M	SA	1	26/02/09	551	Captivity
M09	M	AD	1	26/02/09	578	Captivity
M25	M	AD	3	06/02/14	604	Captivity
M38	M	AD	4	26/01/15	494	Wild
M40	M	SA	4	19/10/15	350	Wild
M43	M	AD	4	05/10/16	705	Wild
M48	M	SA	4	21/11/17	280	Wild

RS, reintroduction site; RD, reintroduction date; RtDs, radiotracking days; AD, adult deer; SA subadult deer.

**Table 2 animals-12-00980-t002:** Estimated total home range and maximum distance from release site for 32 GPS-tracked Corsican red deer.

Deer	MCP 95% ha	BBMM 95% ha	BBMM Core ha	Distance m
M01	1198.00	985.38	353.89	4145.39
F01	1456.14	608.04	194.20	4901.29
F02	1652.16	1092.61	366.77	5186.05
F04	1620.34	1082.27	399.19	4658.35
F05	1467.32	773.39	239.00	5300.74
F07	71.09	94.13	42.81	1923.29
F08	85.00	120.46	49.38	3487.63
F12	1337.00	1308.25	515.53	4349.33
F13	551.10	240.28	92.05	3920.92
F14	1554.32	994.45	373.28	4399.98
F17	1505.09	1517.29	659.45	4925.00
F23	65.23	73.23	27.00	1358.76
F24	74.08	131.10	57.33	1948.60
F30	453.12	796.77	204.58	2724.17
F41	294.79	360.09	179.03	3539.33
F42	302.29	360.49	165.52	2071.96
F43	407.75	1223.86	157.26	1676.28
F54	128.71	183.93	76.88	2654.39
F55	2534.08	1086.20	373.74	5709.06
F56	2418.75	872.86	376.55	9130.35
F64	750.10	604.03	178.14	4394.00
M02	1456.56	1098.90	417.62	4795.08
M04	1147.55	1259.54	472.60	4577.46
M06	191.04	237.73	87.90	3485.01
M07	528.67	504.91	157.73	5402.08
M08	592.51	2078.35	492.35	2842.40
M09	563.05	386.80	138.47	3466.05
M25	287.01	284.59	106.32	2331.00
M38	15,985.53	4257.88	956.20	22,201.29
M40	1583.92	2082.93	881.29	15,843.96
M43	5057.67	5498.64	1903.89	8067.77
M48	284.16	373.74	143.20	2399.46

**Table 3 animals-12-00980-t003:** Differences in home range size and distance from the release point.

Variable	MCP 95%	BBMM 95%	BBMM Core	Distance
Sex	W = 105 *p* = 0.5778	W = 77 *p* = 0.09854	W = 83 *p* = 0.1576	W = 95 *p* = 0.3456
Age class	W = 93 *p* = 0.3886	W = 95 *p* = 0.4335	W = 93 *p* = 0.3886	W = 86 *p* = 0.2544
Release Site df = 3	KW = 16.176 *p* = 0.0010 *	KW = 8.396 *p* = 0.0385 *	KW = 11.02 *p* = 0.0116 *	KW = 15.94 *p* = 0.0011 *
Origin	W = 90 *p* = 0.1596	W = 86 *p* = 0.1188	W = 84 *p* = 0.1016	W = 104 *p* = 0.3809

W, Wilcoxon test; KW, Kruskas–Wallis test; *p*, probability value; * statistical significance; df, degree of freedom.

**Table 4 animals-12-00980-t004:** Relative contributions of the environmental variables to the MaxEnt model.

Variable	Percent Contribution	Permutation Importance
Distance form urban areas	45.7	33.4
Distance from core forest areas	21.1	20.8
Distance from principal roads	12.3	12
Distance from forest areas	9	17.9
Acclivity	4	3
Ecological variables	3	3.4
Diversity index	2.4	6.6
Forest fragmentation	1.5	0.4
Exposition	0.7	0.9
Distance from water streams	0.3	1.5

## Data Availability

Databases of GPS locations of tracked deer are available at Agenzia Forestas and at ISPRA; results of sanitary analyses are available at Agenzia Forestas.

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
