# Peer review of "Reintroductions of the Corsican Red Deer (Cervus elaphus corsicanus): Conservation Projects and Sanitary Risk"

_animals, 2022, doi:10.3390/ani12080980_

Round 1

Reviewer 1 Report

Please, find enclosed the corrected manuscript. Sincerely

Author Response

fr

Reviewer 2 Report

Dear Authors, the paper, topic, telemetric research and results are very interesting, but some things need to be improved. I have also suggestion to ask a native speaker to check your paper. I saw - in my opinion - many English mistakes. I will describe everything line by line:

2  - the title: I don't like "Artificial translocations" -  maybe better would as you write later: Reintroductions? It is logical that these are not natural, by with made by human hand...

8 - "3" as upper index

11 - without the 

13 - contest or context?

15  - without the 

17 - maybe: on the base instead of on the baseline

17 - obtained results, without the

17 - to build instead of build

18 - to pann

21 - I think a word "too" doesn't match here

28-29 - Based on the obtained results....

29  - a Species Distribution Model (SDM) - big letters

30  - from the union - you should find more pooper synonym

36 will be instead of were? English native speaker should check it.

36 - translocating or translocated?

37 - Species Distribution Model

41 - without the

47-48, 50 - put full Latin names e.g. Dama mesopotamica (Brooke, 1875), etc.

49 - are you sure few or a few? There is a difference in meaning

59 - this is a first place where you can put a full Latin name of described species in whole text

64 - Directives

70  - 1950 year

83 - deer (as cattle and goats) - first both singular or plural, decide, second: are you sure of word "as" ? Maybe "like"? "similar to"?

87 - without the

90 - home ranges and movements

91 - ask a Native speaker - plural is deers. Be sure of this and change in whole text if nessesary.

101 - without the, general deer (the same like animals)

102- withoout the, avoiding capture...

108/109 - small letter: adult, subadult, yearings. I would rather start from the opposite: first yearings, then subadult, adult. It is more logical order. 

134 - Figure 1. Sign these two figures as A (Italy) and B (Sardinia). Then, on figure 1A add contures of neughbouring countries to Italy. It doesn't look professional right now.

Change Kilometres to km next to scale and quit from 4,5 and 9. Make a scale 0-20, not 0-18. 

Why sites on this map are sifferent in size? Probably these are countours from telemetric study? There are big differences in size. Maybe you should put borders of areaa that you called site1, 2, 3 and 4. Or maybe only change a signature to this figure. 

On each map I would change z background. It would be more interesting to see e.g. corine land cover. Or maybe on this figure you can leave as it is, but on the other figures change the background. 

141 - Loteck or Lotek?

142-143 - can you explain why there is a difference between different numbers of localizations per day? Site 1 and 2 - six, whereas site 3 and 4 - only two. The next sentence doesn't explain this - it is not the same mean locations per day, that's why Im asking about it.

155 - MCP instead of MPC

156 - check the name - is it correct? "utilizing adehabitatHR Package"

167-168 try to thicken your text in Microsoft Word

175 - maybe it should sound like "....presence data of free living animals in south Sardinia"

instead of "....presence data of animals living in south Sardinia natural populations"

183 - (20 m) instead of (20m)

189 - probably there is a big mistake in English meaing in the text. In my opinion everywhere should be forest instead of wood.... so it should be rather "we used forest fragmentation classes" etc. everywhere. Be careful of it. 

200 -  '94 captured deer' instead of '94 deer captured'

200 - would be or were? It is past. It is done.

207 - "Of them 32, were used...." - it is not a proper construction of the sentence. E.g. Thirty-two of them were used....

216 - fifth column in the table - I would write Distance instead of Dist, it is not longer than names of other

229 - by the union of.... it is not a proper word. Maybe:  "by connection"? 

233 - Figure 3 - please change a background, as I suggested before.  The not The in figure caption. Selected, not - the selected. The same scale like in previous figure.

239 - Figure 4 - put a frame to figure. Change Kilometres into km. On background of this data I would put data from Figure 3. It would be more interesting and allow you to delete figure 3. 

241-242 - put a break between lines.

244, 246 - problem of "wood" words (and inside Table 4), like explained before.

255 - sitr 4 instead of area 4? You should use one nomenclature e.g. sites, not areas.

261 - if you discuss your results with authors that made MCP 100% analysis it is a pity, that you did not the same, but OK. Remember for next papers. 

265-267 - it is good that you underline it, I would not compare  home ranges of European red deer, because of continuity of ecosystems. For them island would be an ecological trap. 

270 - new instead of novel

274 - maybe just 1,4 km - 22 km instead of 1,358.76 m to 22,201.29 m.

275 - .... in the study area 4? Maybe "... in the site 4 called .... "

277 - without the , as well as don't write single animals - it does not sounds well. Rather "individuals", we know that animals.

278 - was, not is

283, 284 - wood  case again.

290 - "in more quick time" -  it is not correct expressions. Maybe: "quciker" ?

293-294 - "with the need to limit  spreading of BTV, especially among domestic sheep"

instead of  "with the need to limit the spread of BTV and the spread of the disease to domestic sheep"

296 - one bracket [61,62,27]

301-302 -  as predictors of the BTW spreading in Sardinia [64] 

303 - Figure 5 - remarks concerning a scale are the same for each figure. The grey colour is not so good visible...  Maybe use a transparency option of the background layer?

304 - ...municipality...

307 - Conservation of endangered taxa could play a significant role in their conservation - it need to be re-write

319  - actual, not the actual

321 - maybe better version will be: "... of a widespread of wild population or free living population of Corsican red deer..." instead of "...of a widespread natural Corsican red deer population..."

330, 331 - shorter pauza

334-335 - please delete this sentence:  Please turn to the CRediT taxonomy for the term explanation. Authorship must be 334
limited to those who have contributed substantially to the work reported.

350 - without ( ), just 80. In 3 and 4 positions of references you put number after year, here not. Do the same accroding to MDPI requirements. 

394  - Ital. instead of Ital?

479 - without ; in the end 

Important remark ro References chapter: check all positions and add DOI number where is possible. Put it according do MDPI reqirements:

e.g. 34-36. DOI:  10.1017/S0030605307012069.

Good luck

Author Response

Francesco Riga
